# Vulvar Cancer: 2021 Revised FIGO Staging System and the Role of Imaging

**DOI:** 10.3390/cancers14092264

**Published:** 2022-04-30

**Authors:** Mayur Virarkar, Sai Swarupa Vulasala, Taher Daoud, Sanaz Javadi, Chandana Lall, Priya Bhosale

**Affiliations:** 1Department of Diagnostic Radiology, University of Florida College of Medicine, 655 West 8th Street, C90, 2nd Floor, Clinical Center, Jacksonville, FL 32209, USA; mayur.virarkar@jax.ufl.edu (M.V.); chandana.lall@jax.ufl.edu (C.L.); 2Department of Diagnostic Radiology, The University of Texas MD Anderson Cancer Center, 1515 Holcombe Blvd., Houston, TX 77030, USA; tedaoud@mdanderson.org (T.D.); sanaz.javadi@mdanderson.org (S.J.); priya.bhosale@mdanderson.org (P.B.)

**Keywords:** 2021 vulvar cancer staging, FIGO staging, imaging of vulvar cancer

## Abstract

**Simple Summary:**

Primary vulvar malignancy is a rare gynecological neoplasm constituting 5–8% of cases. Considering the prognostic capability of staging, FIGO has introduced the revised 2021 staging for vulvar cancer. We review the etiopathogenesis revised 2021 International Federation of Gynecology and Obstetrics (FIGO) classification and emphasize imaging in the staging of vulvar cancer.

**Abstract:**

Vulvar cancer is a rare gynecological malignancy. It constitutes 5–8% of all gynecologic neoplasms, and squamous cell carcinoma is the most common variant. This article aims to review the etiopathogenesis revised 2021 International Federation of Gynecology and Obstetrics (FIGO) classification and emphasize imaging in the staging of vulvar cancer. The staging has been regulated by FIGO since 1969 and is subjected to multiple revisions. Previous 2009 FIGO classification is limited by the prognostic capability, which prompted the 2021 revisions and issue of a new FIGO classification. Although vulvar cancer can be visualized clinically, imaging plays a crucial role in the staging of the tumor, assessing the tumor extent, and planning the management. In addition, sentinel lymph node biopsy facilitates the histopathological staging of the draining lymph node, thus enabling early detection of tumor metastases and better survival rates.

## 1. Introduction

Primary vulvar malignancy is a rare gynecological neoplasm constituting 5–8% of cases [1]. It is the fourth most common gynecological malignancy that usually affects post-menopausal women with a median age of 68 years [2,3]. The American Cancer Society estimates that around 6330 vulvar cancers will be diagnosed and approximately 1560 vulvar cancer deaths will occur in 2022. Squamous cell carcinoma (SCC) constitutes greater than 90% of vulvar cancer cases, followed by melanoma, adenocarcinoma, basal cell carcinoma, sarcoma, and undifferentiated type [4]. Vulvar SCC (VSCC) is characterized by morphological variants, including keratinizing, basaloid, warty, and verrucous types. For the last ten years, there has been a 0.6% increase in the incidence of vulvar cancer per year and a 1.2% increase in mortality rates [4,5]. The increase in 5-yearly average incidence was higher in women < 60 years when compared to women of all ages (11.6% vs. 4.6%) [6]. The age-standardized incidence is 1.4 per 100,000 women between the years 2003 and 2007.

Patients with vulvar cancer may be asymptomatic or present with pruritus, bloody discharge, palpable mass, or pain [2]. Clinically, the lesion appears as thickened or discolored skin or a flat, raised, ulcerated, or plaque-like lesion [2]. Around 59% of vulvar cancers demonstrate localized disease, whereas 30% and 6% metastasize to regional lymph nodes and distant sites, respectively [2,7]. The 5-year survival rate is 86%, 53%, and 19% for localized disease, regional spread, and distant spread, respectively [7].

## 2. Background

The vulva is a diamond-shaped structure comprising mons pubis, genitocrural folds, labia majora and minora, clitoris, vestibule, and Skene and Bartholin vestibular glands (Figure 1) [8]. The skin of the labia is the most common vulvar area associated with cancer, while the clitoris and vestibular glands constitute the rare zones of the cancer [2,9]. Nearly 70% of vulvar cancers involve labia majora and minora, and 15–20% of cases involve the clitoris [7]. It is critical to be familiar with the lymphatic drainage of the vulva for lymph node staging. The lymph nodes are drained primarily through three groups of superficial inguinal lymph nodes: (i) medial group: medial to the femoral and greater saphenous vein; (ii) intermediate group: near the femoral and saphenous vein; and (iii) lateral group: near lateral one-third of the groin [1]. Then the lymphatics traverse the cribriform fascia to enter deep inguinal lymph nodes and later into paraaortic lymph nodes. Occasionally clitoris lymph drainage can bypass superficial inguinal lymph nodes and drain directly into deep inguinal lymph nodes or rarely into external iliac lymph nodes [4].

The vulvar SCC can be divided into human papillomavirus (HPV)-dependent and HPV-independent subtypes based on etiopathogenesis. Around 43% of vulvar cancers are secondary to HPV [5]. HPV 16 (77%) followed by HPV 33 (10.6%) and HPV 18 (3%) are the most common strains associated with VSCC [10]. For either subtype of VSCC, vulvar intraepithelial neoplasia (VIN) is the precursor lesion defined by the histopathological atypia. VIN is classified into usual VIN (uVIN) (also termed as high-grade squamous intraepithelial lesion (HSIL)) and differentiated type VIN (dVIN) [6]. The uVIN is associated with chronic HPV infection, whereas dVIN is HPV-independent and is associated with a chronic inflammatory condition, lichen sclerosis. Women with lichen sclerosis have a relative risk of 38.4 for the evolution of dVIN-associated VSCC [11]. It is critical to distinguish between uVIN and dVIN as the latter is associated with a higher risk of malignant transformation (5.7% vs. 32.8%) [12]. Genomic characterization of the lesion, as described in other sections, aids in differentiation among uVIN and dVIN. Furthermore, advancing age increases the risk of dVIN progression to the cancer [6]. Usually, uVIN lesions transform into warty or basaloid variant SCC and affect younger women, constituting one-third of cases [13,14]. The dVIN lesions are the precursor types for keratinizing variant SCC and commonly affect older women.

## 3. Genomics of Vulvar Cancer

Although 80% of HSIL cases are associated with HPV DNA, only 1.5% and 28.6% of dVIN and VSCC cases are HPV-positive [10]. HPV-positive vulvar cancer is characterized by the over-expression of E6 and E7 viral oncoproteins, which inactivate p53 and pRB (retinoblastoma transcriptional corepressor 1) protein. E6 oncoprotein degrades p53 by tagging it with E6-associated protein, a ubiquitin-protein ligase. E7 oncoprotein degrades pRB, releasing cellular E2F transcription factors responsible for cell-cycle activation. Consequently, cyclin-dependent kinase (CDK) inhibitors, including p16, p21, and p14, upregulate, leading to the downregulation of cyclinD1 and its complex formation with CDK4 [15]. Of all the three CDK inhibitors, p16 is reliable in HSIL and can be used as a surrogate marker for HPV infection [14]. It has a sensitivity and specificity of 100% and 98%, respectively, in the classification of VSCC into HPV-dependent and -independent types [16]. The p16 is a tumor suppressor protein encoded by the CDKN2A gene. Normal cells arrest the cell cycle by complexing with CDK 4 and 6 and attenuating pRB phosphorylation, which is required for E2F release. HPV leads to over-expression of p16, resulting in continuous “block-positive” nuclear or nuclear and cytoplasmic staining extending from basal to superficial layer on the immunohistochemistry [6,15]. Lee et al. reported higher 5-year progression-free survival (65% vs. 16%) and overall survival (65% vs. 22%) in patients with p16-positive vulvar cancer when compared to the p16-negative disease [17].

In contrast to HPV-dependent vulvar cancers, somatic TP53 mutation is implicated in 30–80% of the dVIN [18]. Nearly 58% of mutations are C: G to T: A transition types due to hydrolytic deamination of 5-methylcytosine residues [19]. The dVIN shows solid nuclear staining on immunohistochemistry due to the accumulation of hyperstable and mutant p53 protein in the basal layer (7). The TP53 gene encodes p53 protein, a multifunctional transcription factor and a tumor suppressor. The p53 protein regulates cell-cycle progression, DNA integrity, and apoptosis and prevents genomic mutations. Hence, it is termed “the guardian of the genome.” Mutations in the TP53 gene lead to loss of tumor-suppressor activity of p53. The presence of TP53 mutation is associated with poor survival and short disease-free intervals compared to HPV-dependent SCC. Many chemotherapeutic agents act through the p53 pathway [20], and loss of the p53 tumor suppressor gene can result in chemoresistance and radio-resistance.

## 4. Role of Imaging in Vulvar Cancer

Although imaging is not an integral part of the vulvar cancer FIGO staging, it provides crucial information on the status of tumor extension to deeper tissues, lymph nodes, and distant organs. Due to its superior tissue resolution, MRI is the modality of choice to assess the vulvar anatomy. The normal vulva is hypo- to isointense on T1 weighted imaging (T1WI) and hyperintense on T2 weighted imaging (T2WI) sequences [21]. The vestibular bulb and clitoral units are T2 hyperintense compared to muscle and exhibit avid enhancement on contrast administration. On imaging, vestibular bulbs appear as a U-shaped structure surrounding the lower vagina and urethra, whereas the clitoral appears as a curved structure. Compared to pelvic muscles, the bulbospongiosus and ischiocavernosus muscles are thin, and the hypointensity can be observed just underneath the labia majora and skin. The urethral complex has an outer hypointense and inner hyperintense layer, giving the target-like appearance to axial T2WI and contrast-enhanced T1WI sequences.

According to the European Society of Urogenital Radiology, a minimum field strength of 1.5 T is recommended in vulvar cancer evaluation. Based on MRI findings, around 83% of the vulvar lesions can be accurately classified [22]. Patients are advised to fast for at least 4–6 h before the imaging, void the bladder, and are prescribed anti-peristaltic agents to limit the bowel movements. Vaginal distention with ultrasound gel aids in better identification of smaller vulvar tumors with or without vaginal infiltration. The patients should be supine and imaged with an eight-channel cardiac or a phased array pelvic coil and get the intravaginal gel. The MRI pelvic protocol and MRI-focused report for vulvar imaging are delineated in Table 1 and Table 2. They include axial T1WI, T2WI, fat-suppressed T1WI (FS-T1WI), and gadolinium-enhanced FS-T1WI. Axial diffusion-weighted imaging (DWI), with a similar angle to T2WI, adds value for a better tumor definition. Vulvar cancer is appreciated better on FS-T2WI than non-fat-suppressed sequences. The former can be explained by the high amount of fat in the perineal region which gets suppressed on fat-saturation images, improving the lesion conspicuity [23,24]. Axial T1WI and T2WI sequences with a large field of vision from aortic bifurcation to below the vulva determine the extent of the tumor alongside lymph node and pelvic bone metastases. Gadolinium-enhanced and axial 3D dynamic fat-suppressed T1WI sequences and a sagittal fat-suppressed T1 allow the assessment of tumor characterization and extent. The 3D dynamic images are acquired at a contrast resolution of 13–16 s for 3 to 5 min. Contrast-enhanced MRI assists in the visualization of smaller tumors and the involved adjacent organs such as the vagina, urethra, and anus. In a study by Kataoka et al., contrast administration improved tumor detection accuracy from 75% to 85% [22,23]. In addition to T1WI and T2WI sequences, axial FS-T2WI sequence and 3D dynamic gadolinium-enhanced axial T1WI sequence demonstrate and differentiate the recurrence of malignancy from radiation-induced fibrosis [21].

## 5. International Federation of Gynecology and Obstetrics Staging of Vulvar Cancer

The International Federation of Gynecology and Obstetrics (FIGO) has determined the staging of vulvar cancer since 1969 [25]. In 1988, FIGO classification surgical and pathological characteristics alone were considered in staging vulvar cancer. Later, in 2009, the revisions were made to include the tumor extension to adjacent structures and the number and extent of lymph node metastases. Considering the prognostic capability of staging, FIGO has introduced the revised 2021 staging for vulvar cancer (Table 3 and Figure 2). Imaging had no role in FIGO 2009 classification. However, the 2021 revision incorporated cross-sectional imaging findings into vulvar cancer staging, and this staging applies to all types of vulvar cancer except vulvar melanoma.

### 5.1. Stage I

The stage I vulvar cancer classification remained unchanged from the 2009 FIGO staging except for the depth of invasion definition. In the 2009 classification, the depth of invasion is measured from the epithelial-stromal junction of the adjacent, most superficial dermal papilla to the deepest extent of tumor invasion. However, it is redefined in 2021 as a measurement from the basement membrane of the adjacent, deepest, dysplastic, tumor-free rete ridge to the deepest extent of the invasion [13]. Stage I cancer signifies the tumor confined to the vulva. It is sub-classified into stage IA and IB disease based on tumor size and stromal invasion. In stage IA, the tumor measures ≤ 2 cm in size and invades stroma ≤ 1 mm from the basement membrane of the deepest tumor-free or dysplastic rete ridge. Stage IB is defined by the tumor > 2 cm size and >1 mm stromal invasion (Figure 3 and Figure 4). There is no adjacent structural, nodal, or distant organ involvement in stage I disease. The role of imaging in stage I disease is debatable as the tumor can be certainly evaluated by physical examination. Magnetic resonance imaging (MRI) is the imaging modality of choice in tumors > 2 cm, not confined to vulva and perineum, and >1 mm stromal invasion [24]. There is limited literature on the efficacy of MRI in tumors without the former characteristics [24]. Compared to muscles, the vulvar tumor appears as a solid hypo- to isointense on T1WI and intermediate- to hyperintense (“evil gray”) mass on T2WI sequences [23,26]. Hyperintensity on T2WI imaging sequences is due to internal necrosis in the case of larger tumors. Tumor demonstrates early arterial enhancement on dynamic contrast MRI and restricted diffusion on DWI sequences [23,26].

### 5.2. Stage II

Stage II represents any sized tumor extending to the lower one-third of the urethra, vagina, or anus without metastases to lymph nodes. There are no additional revisions in this stage from FIGO 2009 classification (Table 4 and Figure 5). In addition to the size of the tumor, MRI assesses the extension of locally advanced vulvar cancer to superficial (labia majora and minora and clitoris) and deep structures (urethra, vagina, and anus). Clinical examination alone is challenging in patients with urethral involvement and absent clinical findings. In such cases, disruption of the target appearance of the urethra on T2WI and contrast-enhanced T1WI is suggestive of a urethral infiltration [26].

In contrast, the absence of disruption on MRI reduces the likelihood of urethral infiltration. In patients with vaginal wall involvement, the disruption of low signal intensity of the vaginal wall on T2WI and contrast-enhanced T1WI may be observed [26]. Similarly, the interruption of the low-signal intensity of the anal sphincter on T2WI and the muscles on post-contrast T1WI indicates anal wall involvement [24]. In case of uncertainty, endo-anal ultrasound can be performed to delineate muscle layers better. DWI aids in assessing the tumor extension by demonstrating the continuation of restricted diffusion into the urethral wall, vaginal wall, or anal sphincter [26].

### 5.3. Stage III

Stage III of FIGO 2021 classification is divided into three substages A, B, and C (Figure 6). It includes vulvar tumors of any size extending to the upper part of adjacent structures or any number of nonulcerated and nonfixed lymph nodes. This contrasts with the FIGO 2009 classification, where tumor extension to upper parts of adjacent structures is now considered stage IVA (Table 5) [13,24]. Stage IIIA contains tumors of any size with extension to the upper two-thirds of the urethra, bladder mucosa, upper two-thirds of the vagina, rectal mucosa, or regional inguinofemoral lymph node metastases ≤ 5 mm. Stage IIIB is defined as tumor metastasis > 5 mm to regional lymph nodes. Stage IIIC contains tumors with regional lymph node metastases with extracapsular spread. The extracapsular spread is associated with a worse 5-year survival rate than patients with intranodal metastases alone (34% vs. 66%) [24].

Rectal mucosa, bladder mucosa, vagina, and urethral infiltration can be identified by visualizing the intermediate signal intensity of the vulvar tumor on T2WI. Multiplanar imaging is recommended for the accurate assessment of adjacent organ involvement. For instance, the sagittal plane is more beneficial in evaluating the caudo-cranial extension of the tumor through the fornices to the upper portion of the vagina, the anal sphincter to the rectum, and the urethra to the base of the bladder. Axial and coronal planes assess the lateral tumoral extension to the pelvic bones and muscles.

Lymph node metastases are the most important prognostic factor, if present, resulting in the reduction in the 5-year survival rate from 95% to 62% [27]. The 5-year survival rate drops from 80% in node-negative patients to 50% in node-positive patients and 12% in patients with >4 positive lymph nodes [28]. Around 88% of node-negative patients are free from disease in two years, whereas 60%, 43%, and 29% of patients with one, two, or ≥3 lymph nodes, respectively, are free from the disease within two years [7]. Inguinofemoral nodes are considered regional lymph nodes and the superficial medial group of inguinal lymph nodes in the sentinel nodes in a large percentage of vulvar carcinoma [23]. Lateral vulvar lesions usually drain to ipsilateral lymph nodes, whereas the lesions within 1 cm of the midline may drain bilaterally [26]. Moreover, the tumor involving adjacent structures such as the vagina, urethra, or anus (above the dentate line) may spread directly to the pelvic lymph nodes.

The relevance of ultrasound in lymph node evaluation has been reported in several studies. It has a sensitivity and specificity ranging from 76 to 100% and 69 to 91%, respectively. This wide range can be explained by the several imaging features of lymph nodes such as short-axis diameter, long-axis to short-axis ratio, shape, fatty hilum, and vascularization. The increased peripheral vascularity and a spectral waveform with high resistance suggest metastatic lymph nodes. Ultrasound combined with fine-needle aspiration cytology can avoid unnecessary lymph node dissection. Hall et al. reported that the combination has 93% and 100% sensitivity and specificity, respectively [29]. However, the limitations of ultrasound include operator dependency and sampling error in patients with nodal micro-metastases [30]

Lymph node involvement can be suspected on MRI based on several criteria, including short-axis diameter > 1 cm, rounded morphology with long-axis to short-axis ratio < 1.3:1, cystic nodal change, loss of fatty hilum, and irregular contour [23,26]. Evaluation based on short-axis diameter > 10 mm has 89% and 91% sensitivity and specificity, respectively [23]. The long-to-short-axis ratio is most accurate (85%), the presence of necrosis is most specific (93%), and readers’ confidence in lymph node metastases is most sensitive (87%) to estimate tumor staging [1]. Ultrasmall superparamagnetic iron oxide (USPIO) contrast agents are the nanoparticles used to detect lymph node involvement on MRI. USPIO particles are incorporated into reticuloendothelial cells such as macrophages. On MRI, the accumulation of USPIO particles in the normal lymph nodes results in low signal intensity on T1WI and T2WI sequences. However, when the nodal reticuloendothelial cells are replaced by tumor cells, there is no uptake of USPIO particles and no signal change. In a recent meta-analysis of various tumors, including vulvar cancer, the incorporation of nanoparticles demonstrated a sensitivity and specificity of 88% and 96%, respectively [31].

CT is a commonly used preoperative assessment tool to evaluate nodal and extra-nodal spread of the tumor. CT aids in assessing the proximity of metastatic inguinal lymph nodes to the blood vessels, evaluating the pelvic lymph nodes, or planning postoperative management in patients with surgically proven inguinal metastases (8). The GROINSS-VII study recommended CT as a preferred imaging modality to exclude lymph nodal metastases before SLN biopsy. This has been questioned by Pounds et al., who reported the sensitivity and specificity of CT as 59% and 78%, respectively, in the evaluation of inguinal lymph nodes. In their study, over one-third of the patients with negative nodes on imaging had inguinal metastases (35).

### 5.4. Stage IV

Stage IV comprises tumors of any size fixed to the bone or with the presence of fixed ulcerated lymph node metastases or distant structural metastases (Table 6). Stage IV is subdivided into IVA and IVB. Stage IVA describes the characteristics of metastases manifesting as fixed or ulcerated lymph nodes or tumors fixed to the pelvic bone. Stage IVB signifies distant metastases (Figure 7). Pelvic lymph node involvement is considered distant metastases and is also categorized under FIGO stage IV [26,32]. The Cloquet node is an essential indicator of pelvic nodal spread as it is the lowest external iliac node located at the entrance of the femoral canal. Distant metastasis is rare and often preceded by local tumor recurrences. Lungs, liver, bone, lymph nodes, and skin are most involved. The prognosis is poor in patients with distant metastases, with a 2-year survival rate of 11.3% and a median survival rate from a diagnosis of 5.6 months [24].

According to National Comprehensive Cancer Network guidelines, the PET/CT with 18-fluorodeoxyglucose is recommended for patients with suspected metastases [33,34]. It is a non-invasive tool for pre-operative assessment and staging of vulvar cancer. The ^18^F-FDG PET/CT detects primary vulvar lesions with 100% accuracy. However, its sensitivity (50–100%) and specificity (67–100%) ranges are vast, making its application less feasible in nodal staging. The overall sensitivity could be attributed to its limited ability to identify lymph node metastases < 5 mm [7,33]. PET/CT scans exhibit false-positive findings in the presence of inflammation and false-negative results in necrotic tissue. Triumbari et al. reported that PET/CT aids in excluding groin metastases in vulvar cancer patients with a sensitivity and specificity of 70% and 90%, respectively [35]. FDG PET/CT can be supportive, in cases of suspected pelvic nodal or distant metastases, given its 100% negative predictive value [7]. The mean SUV_max_ for metastatic lymph nodes ranges from 6.1 to 11.0 [7]. PET/CT also assists in evaluating the treatment response and detecting the disease recurrence and hence is considered superior to conventional imaging in this perspective. Albano et al. reported the sensitivity, specificity, and accuracy of PET/CT in detecting recurrent disease as 100%, 92%, and 98%, respectively [36]. Early detection of recurrence aids to stratify the patients better and begin prompt therapies to improve outcomes and survival.

## 6. Treatment

As the tumor margin status has been accepted as a significant prognostic factor for tumor recurrence, radical local excision with at least a 1–2 cm margin is recommended in early-stage vulvar cancer (Figure 8, Figure 9 and Figure 10) [37]. Margins can be reduced if the tumor is close to midline structures such as the clitoris, anus, or urethra to preserve their function. In case of positive margins (<8 mm from the tumor), re-excision, or, if unresectable or involving the urethra, anus, or vagina, adjuvant external beam radiotherapy is recommended [38]. The total radiation dose shall be between 60 Gy and 70 Gy [39]. Inguinofemoral lymphadenectomy (IFLD) is not recommended due to the low risk (<1%) of lymph node metastases in stage IA disease. Patients with stages IB or greater possess a 20–30% risk of lymphatic metastases; hence, unilateral lymphadenectomy or sentinel lymph node (SLN) biopsy should be performed if the tumor is <4 cm and located ≥ 2 cm from the midline [4]. Compared to IFLD, SLN has a sensitivity of 92% and NPV of 97–98% in identifying metastases [40]. For a tumor within 2 cm of the vulvar midline, bilateral IFLD or SLN biopsy is recommended [4,5,41]. Contralateral lymphadenectomy is advised for patients with positive lymph node metastases or undetectable SLN [4,5]. Positive SLN with <2 mm metastases can be treated with radiotherapy, sparing from groin lymph node dissection [24]. The GROINSS-V trial reported that SLN metastases ≤ 2 mm have better disease-specific survival rates than SLN metastases > 2 mm (94% vs. 70%; *p* = 0.001) [42].

SLN mapping was introduced in 1977 by Cabanas and in 1994 for the first time in the vulvar cancer [23,43]. Previously, radical surgery with IFLD was the standard treatment for early-stage vulvar cancer. Although radical surgery improved the 5-year survival rates, the morbidity was significant due to the complications such as infections, pain, lymphocele, and chronic lymphedema of the lower extremities. These complications often prolong the duration of hospitalization and delay the adjuvant therapy introduction when needed. Lymphedema is reported in 30–70% of patients undergoing complete IFLD (48). 

Moreover, only 10–26% of the dissected lymph nodes demonstrate metastases, while approximately 80% of patients were reported to be free from lymph node spread [43]. Hence novel diagnostic tests with high sensitivity are required to exclude patients without lymph node metastases. Clinical examination alone is not reliable because 16–24% of patients have normal clinical findings, and 24–41% of clinically enlarged lymph nodes may have normal results in the histological examination [44]. Imaging such as ultrasound, CT, MRI, and PET are reported to have a wide sensitivity range (45–86%), and negative likelihood ratio range (0.12–0.6) and hence are less reliable. The SLN biopsy reduced the postoperative morbidity due to lymphadenectomy without jeopardizing the LN metastases detection.

All the patients with stage III or IVA shall receive radiotherapy with concurrent chemotherapy (Figure 11). The same therapy covering the primary lesion is also recommended in patients with negative IFLD. Patients with residual tumors may be considered for resection, or if unresectable, external beam radiotherapy is recommended [7]. Palliative care and better quality of life are the primary treatment goals in patients with distant metastases. Chemoradiotherapy can be contemplated at the primary tumor site for symptom relief. Incorporating systemic chemotherapy is still controversial in the case of metastatic vulvar cancer, and further studies are necessary to make any recommendations [4]. However, platinum-based chemotherapy regimens alone or combined with another agent such as paclitaxel, vinorelbine, 5-Fluorouracil, or mitomycin C are the most frequently used [45]. Gefitinib and trastuzumab combination regimen has shown efficacy by increasing the radiosensitivity of the vulvar lesion [45,46].

## 7. Surveillance of Vulvar Cancer

Due to a recurrence rate between 12 to 40%, routine surveillance is recommended in all patients of VSCC [4]. The GROningen International Study on SLNs in vulvar cancer (GROINSS-V) study, involving 403 patients, reported that the groin recurrence rate was 2.3% over 3.5 months in patients with negative SLN [47]. Recurrence of vulvar cancer is common in the vulvar and perineal region and usually occurs within two years of clinical presentation [24] and may benefit from tumor restaging with computerized tomography (CT) of the chest, abdomen, and pelvis and shall be managed through radical excision and adjuvant radiotherapy [24]. Among women with positive nodes, the recurrence rate is higher within the first two years when compared to node-negative patients (33% vs. 5.1%) [28]. After two years, the recurrence rates are similar (12%) irrespective of the node status [28]. Patient surveillance should include history and careful clinical examination for vulvar lesions and cervical, perianal, and vaginal neoplasms, all associated with HPV infection [28]. It is essential to emphasize the symptoms of tumor recurrence, including new lesion/mass, vulvar itching, leg or groin pain, lower extremity lymphedema, urinary symptoms, weight loss, and cough. Laboratory studies, imaging, and biopsies are recommended for all suspicious lesions. Surveillance of cancer patients often involves interdisciplinary coordination among specialists and multiple health care providers.

## 8. Conclusions

The revised FIGO classification for vulvar cancer has overcome the limitation of prognostic capability and guides the clinicians in the appropriate management of patients. Incorporating imaging into the classification is a paradigm shift from the older classification. MRI is superior to conventional imaging due to its excellent soft-tissue resolution and aids local vulvar cancer staging by assessing the involvement of adjacent tissues. CT and PET/CT imaging aid in determining the extent of the distant metastases. PET/CT is better in detecting lymph node metastases. As the staging and treatment of vulvar cancer evolve, radiologists must be familiar with the recent FIGO staging of vulvar cancer and help guide management. 

## Figures and Tables

**Figure 1 cancers-14-02264-f001:**
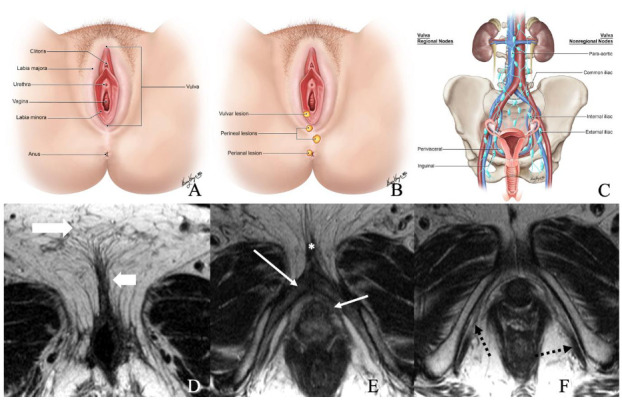
(**A**) Normal anatomy of vulvar region. (**B**) Normal description of location of lesions. (**C**) Regional and non-regional lymph nodes of the vulva. (**D**–**F**) Axial non-fat saturated T2-weighted images (WI) of a 36-year-old-female with normal anatomy. Image D demonstrates labia majora (long thick arrow) and labia minora (short, thin arrow). Image E shows the bulb of the vestibule (short, thin arrow), glans (asterisk), and crus (long thin arrow) of the clitoris. Image F demonstrates the ischiocavernosus muscle (dotted arrow).

**Figure 2 cancers-14-02264-f002:**
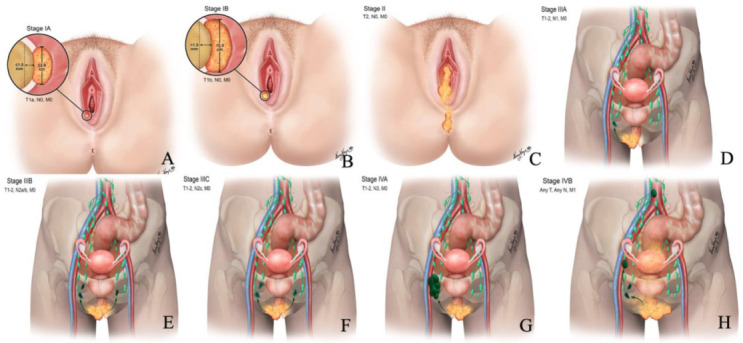
Revised 2021 FIGO staging of vulvar cancer. Stage I tumor confined to the vulva. (**A**) Stage IA tumor size less than equal to 2 cm and stromal invasion less than equal to 1 mm. (**B**) Stage IB tumor size more than 2 cm and stromal invasion more than 1 mm. (**C**) Stage II tumor of any size with extension to lower one-third of the urethra, lower one-third of the vagina, lower one-third of the anus with negative nodes. Stage III tumor of any size with extension to the upper part of adjacent perineal structures or with any number of nonfixed, nonulcerated lymph nodes. (**D**) Stage IIIA tumor of any size with disease extension to the upper two-thirds of the urethra, upper two-thirds of the vagina, bladder mucosa, rectal mucosa, or regional lymph node metastases less than equal to 5 mm. (**E**) Stage IIIB regional lymph node metastases more than 5 mm. (**F**) Stage IIIC regional lymph node metastases with extracapsular spread. Stage IV tumor of any size fixed to bone or fixed, ulcerated lymph node metastases, or distant metastases. (**G**) Stage IVA disease fixed to the pelvic bone or fixed or ulcerated regional lymph node metastases. (**H**) Stage IVB distant metastases.

**Figure 3 cancers-14-02264-f003:**
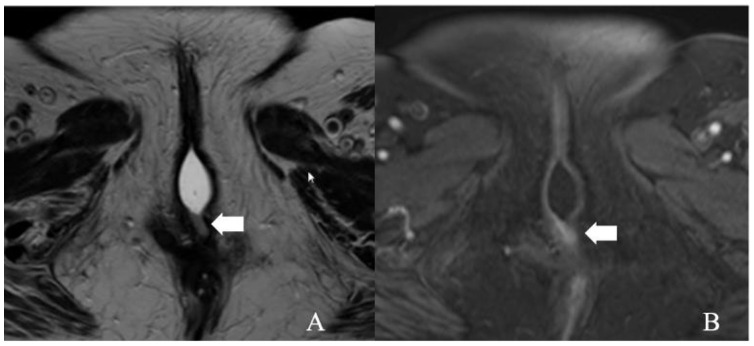
FIGO stage IA. A 43-year-old female with squamous cell carcinoma of the vulva. (**A**) Axial T2 weighted image and (**B**) post-contrast fat-saturated axial T1 weighted MRI image showing an enhancing 1.3 × 0.6 cm T2 intermediate signal biopsy-proven vulvar tumor (arrow).

**Figure 4 cancers-14-02264-f004:**
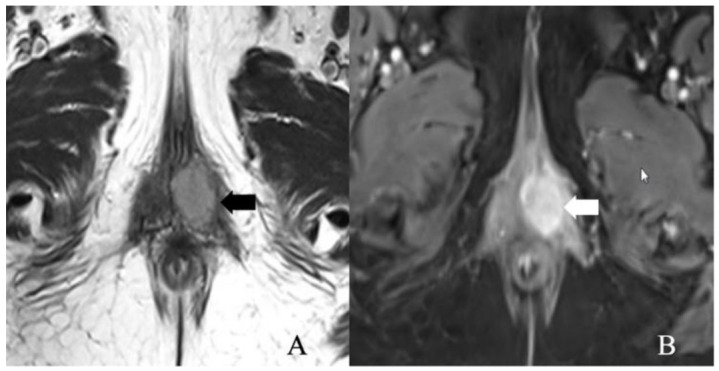
FIGO stage IB. A 52-year-old female with squamous cell carcinoma of the vulva. (**A**) Axial T2 weighted image and (**B**) post-contrast fat-saturated axial T1 weighted MRI image showing an enhancing 3.4 × 2.3 cm T2 intermediate signal biopsy-proven vulvar tumor (arrow).

**Figure 5 cancers-14-02264-f005:**
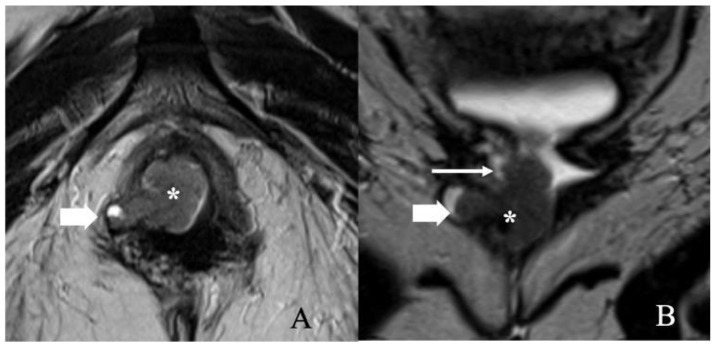
FIGO stage II. A 57-year-old female with squamous cell carcinoma of the vulva. (**A**) Axial T2 weighted image and (**B**) coronal T2 weighted MRI image show a 3.4 × 1.4 cm vulvar tumor (asterisk) involving the right aspect of the lower one-third of the vagina (long thin arrow) and right ischio-anal fossa (short thick arrow). Anteriorly, the mass abuts the lower third of the urethra. No lymphadenopathy was noted. Note that the gel in the vagina helps better delineate the tumor.

**Figure 6 cancers-14-02264-f006:**
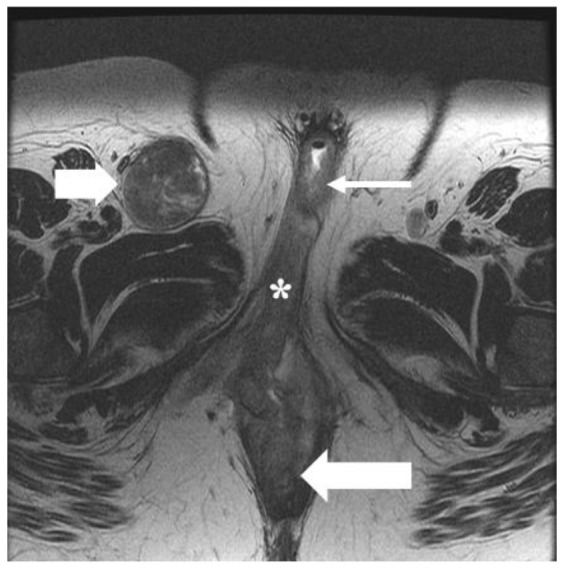
FIGO stage III. A 49-year-old female with squamous cell carcinoma of the vulva. Axial T2 weighted MRI image shows infiltrative vulvar tumor (asterisk) involving, anteriorly, the lower one-third of the urethra (long thin arrow) and, posteriorly, the lower one-third of the anus (long thick arrow). Moreover, there is a metastatic right inguinal lymph node (short thick arrow). The findings correspond to FIGO stage IIIB.

**Figure 7 cancers-14-02264-f007:**
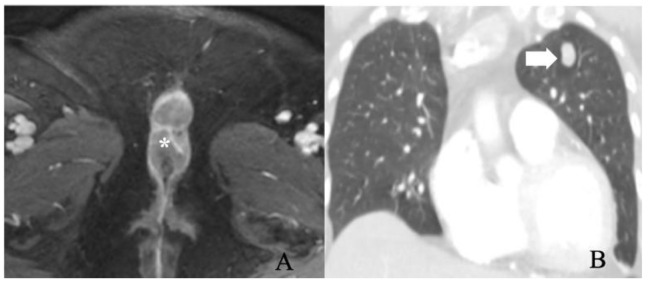
FIGO stage IV. A 56-year-old female with squamous cell carcinoma of the vulva. (**A**) post-contrast fat-saturated axial T1 weighted MRI image shows an enhancing 5.2 × 2.8 cm vulvar tumor (asterisk). (**B**) Coronal lung window contrast-enhanced CT image shows pulmonary nodule (arrow). A biopsy of the nodule was positive for metastasis. The findings correspond to FIGO stage IVB.

**Figure 8 cancers-14-02264-f008:**
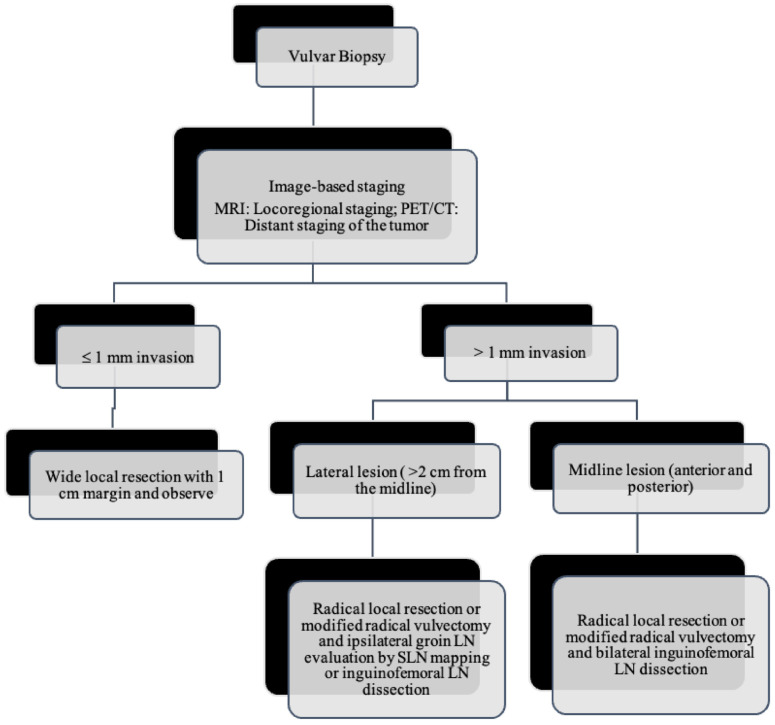
Primary treatment of vulvar cancer.

**Figure 9 cancers-14-02264-f009:**
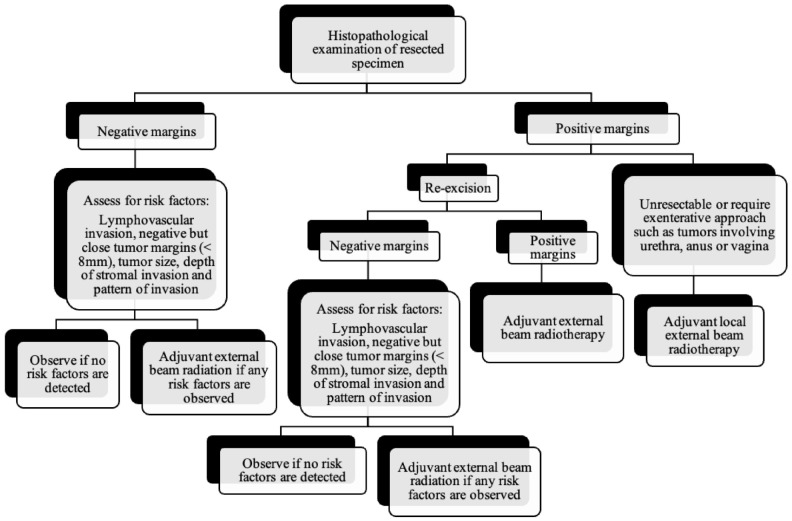
Post-surgical surveillance of vulvar cancer.

**Figure 10 cancers-14-02264-f010:**
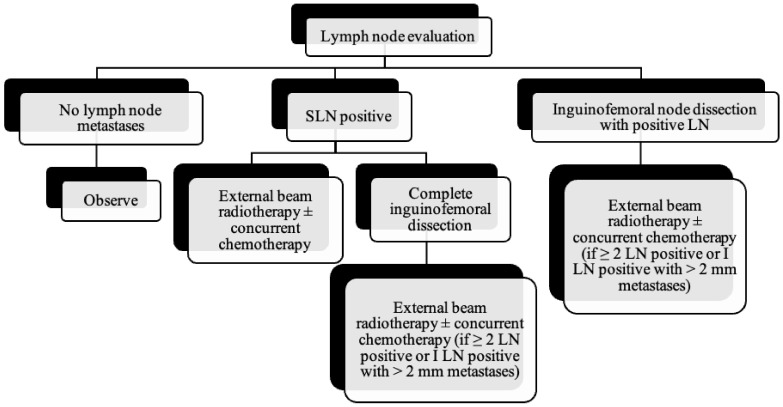
Lymph node evaluation and treatment in patients with vulvar cancer.

**Figure 11 cancers-14-02264-f011:**
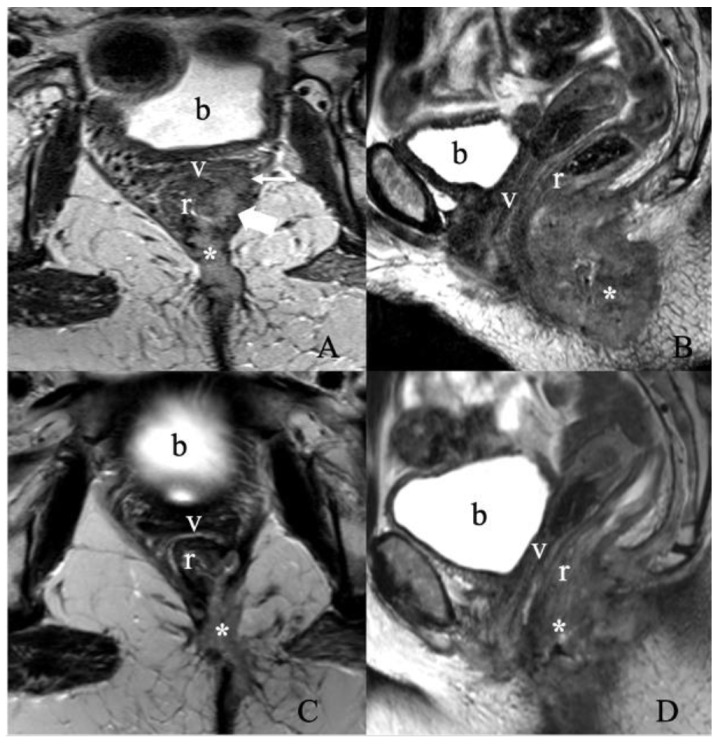
Treatment response assessment. A 55-year-old female with recurrent vulvar squamous cell carcinoma. (**A**) Axial and (**B**) sagittal T2 weighted MRI images show a large vulvar mass (asterisk) invading both the low rectum (thick arrow) and diffusely throughout the left semi circumferential anal canal (not shown), with additional invasion along the left posterolateral aspect of the upper two-thirds of the vagina (thin arrow). No evidence of enlarged lymph nodes. The findings correspond to FIGO stage IIIA. A Follow-up MRI after three months of chemoradiation was obtained. (**C**) Axial and (**D**) sagittal T2 weighted image MRI show an interval decrease in the size of the treated left vulvar mass invading the lower rectum, anal canal, left upper one-third of the vagina, and left lateral pelvic sidewall. (b, bladder; v, vagina; r, rectum).

**Table 1 cancers-14-02264-t001:** Protocol of pelvic MRI.

Series Number	Series Description	FOV	Slice Thickness	Spacing	Frequency Encoding	Freq × Phase
1	Coronal T2 (to include the kidneys)	420	5	0	S/I	288 × 192
2	Sagittal T2	240	5	0	A/P	320 × 224
3	rFOV Sagittal b = 50,600	240	5	0	S/I	96 × 80
4	Axial T2	240	5	0	L/R	320 × 224
5	AxialT1	240	5	0	L/R	320 × 224
6	Axial DWI b = 50,400,800	380+	5	0	L/R	96 × 160
7	Axial 3D Pre	240	5	−2.5	L/R	320 × 224
8	Axial Dynamic 115	240	5	−2.5	S/I	256 × 224
9	+C Sagittal 3D Immediate delay	240	5	−2.5	L/R	256 × 224

All patients receive vaginal gel. Ultrasound gel (60 cc; Aquasonic Clear Ultrasound Gel, Parker Laboratories) is used to distend the vagina. FOV: Field of Vision; DWI: Diffusion-Weighted Imaging.

**Table 2 cancers-14-02264-t002:** MRI-focused report for locally advanced vulvar cancer staging.

MRI Report Findings
Tumor dimension: maximum diameter
Tumor location: lateral/midline/multifocal
Clitoris involvement
Extension to adjacent organs/structures: urethra and/or vagina with caudo-cranial extension specification; lower one-third or upper two-third; urethral meatus; bladder; fourchette area, anus/rectum
Lymph node involvement: inguinofemoral and/or pelvic and/or abdominal
Additional findings: uterus; adnexa; kidneys; and pelvic bones

MRI: Magnetic Resonance Imaging.

**Table 3 cancers-14-02264-t003:** Revised 2021 FIGO staging of vulvar cancer.

Stage	Characteristics
I	Tumor confined to the vulva
	IA—Tumor size ≤ 2 cm and stromal invasion ≤ 1 mm *
	IB—Tumor size > 2 cm or stromal invasion > 1 mm *
II	Tumor of any size with extension to lower one-third of the urethra, lower one-third of the vagina, lower one-third of the anus with negative nodes
III	Tumor of any size with extension to the upper part of adjacent perineal structures or with any number of nonfixed, nonulcerated lymph nodes
	IIIA—Tumor of any size with disease extension to upper two-thirds of the urethra, upper two-thirds of the vagina, bladder mucosa, rectal mucosa, or regional lymph node metastases ≤ 5 mm
	IIIB—Regional ** lymph node metastases > 5 mm
	IIIC—Regional ** lymph node metastases with extracapsular spread
IV	Tumor of any size fixed to the bone or fixed, ulcerated lymph node metastases, or distant metastases
	IVA—Disease fixed to the pelvic bone or fixed or ulcerated regional ** lymph node metastases
	IVB—Distant metastases

* Depth of invasion is measured from the basement membrane of the deepest, adjacent, dysplastic, tumor-free rete ridge (or nearest dysplastic rete peg) to the deepest point of invasion. ** Regional refers to the inguinal and femoral lymph nodes.

**Table 4 cancers-14-02264-t004:** International Federation of Gynecology and Obstetrics (FIGO) stage II definitions.

FIGO Staging System	II
2009	Tumor of any size with extension to the adjacent perineal structures (lower one-third of the urethra, lower one-third of vagina, anus) with no lymph node involvement
2021	Tumor of any size with extension to lower one-third of the urethra, lower one-third of the vagina, lower one-third of the anus with no lymph node involvement

**Table 5 cancers-14-02264-t005:** International Federation of Gynecology and Obstetrics (FIGO) stage III definitions.

FIGO Staging System	III	IIIA	IIIB	IIIC
2009	Tumor of any size, with/without extension to adjacent perineal structures (lower third of urethra, the lower third of vagina, anus) with positive inguinofemoral lymph nodes	With one lymph node metastases (≥5 mm) or with 1–2 lymph node metastases (<5 mm)	With two or more lymph node metastases (≥5 mm) or with three or more lymph node metastases (<5 mm)	With positive nodes with extracapsular spread
2021	Tumor of any size with extension to upper parts of adjacent perineal structures or with any number of nonfixed and nonulcerated lymph nodes	Tumor of any size with disease extension to upper two-thirds of the urethra, upper two-thirds of the vagina, bladder mucosa, or regional lymph node metastases ≤ 5 mm	Regional lymph node (inguinofemoral) metastases >5 mm	Regional lymph node (inguinofemoral) metastases with extracapsular spread

**Table 6 cancers-14-02264-t006:** International Federation of Gynecology and Obstetrics (FIGO) stage IV definitions.

FIGO Staging System	IV	IVA	IVB
2009	Tumor invades adjacent structures or fixed and ulcerated lymph nodes or distant metastases	Upper urethral and/or vaginal mucosa, bladder mucosa, rectal mucosa, or is fixed to the pelvic bone Fixed or ulcerated inguinofemoral lymph nodes	Any distant metastases, including pelvic lymph nodes
2021	Tumor of any size fixed to the bone, or fixed and ulcerated lymph node metastases or distant metastases	Disease fixed to the pelvic bone or fixed or ulcerated regional (inguinofemoral) lymph node metastases	Distant metastases

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
