# Peer review of "Vulvar Cancer: 2021 Revised FIGO Staging System and the Role of Imaging"

_cancers, 2022, doi:10.3390/cancers14092264_

Round 1

Reviewer 1 Report

The review "Vulvar Cancer: 2021 Revised FIGO Staging System and the Role of Imaging" is punctual and timely because it updates almost in real time what are the possibilities of diagnostic integration offered by imaging compared to the latest Revised FIGO Staging System published only in 2021.

I only have a small observation to make:

In the legends of figures 3 and 4, the interpretation of the image indicated as B in the figure is missing.

Author Response

Thank you for reviewing our manuscript. We acknowledged your suggestions and edited the figure 3 & 4 legends.

Reviewer 2 Report

This work is an interesting complement to the works that were published in 2021, including: Vulvar cancer staging: guidelines of the European Society of Urogenital Radiology (ESUR) (O. Nikolić et all) or FIGO staging for carcinoma of the vulva: 2021 revision (A.B. Olawaiye et all).

Some comments:

  1. In Figure 1 (A, B and C), in Figure 2 (A, B, C and D) there are very blurred descriptions (lowercase letters);
  2. In Figures 3 and 4, the descriptions are only marked for figure A, not for Figure B;
  3. In Chapter 5, subsections are incorrectly numbered:
    Line 190: It is 3.1. Stage I should be 5.1. Stage I;
    Line 219: It is 3.1.Stage II and it should be 5.2. Stage II;
    Line 245: It is 3.1. Stage III and a should be 5.3. Stage III;
    Line 319: It is 3.1. Stage IV should be 5.4. Stage IV;
  1. Figure's signatures should be standardized.Once there is, for example, Figure 3 and then Figure-3 (with a line).

Naturally, these comments do not affect the quality of work.

Author Response

Thank you for reviewing our manuscript. We acknowledged your suggestions and edited the manuscript accordingly. 

Reviewer 3 Report

This is an excellent piece of work. The MRI images are useful and the explanations complete.

Author Response

Thank you for reviewing our manuscript. We appreciate your comments.